# Prompting with Pseudo-Code Instructions

**Mayank Mishra,**[*] **Prince Kumar,**[*] **Riyaz Bhat,**
**Rudra Murthy V, Danish Contractor, Srikanth Tamilselvam**
IBM Research AI
{mayank.mishra1,prince.kumar12, riyaz.bhat,danish.contractor}@ibm.com,
{rmurthyv,srikanth.tamilselvam}@in.ibm.com

## Abstract

Prompting with natural language instructions has recently emerged as a popular method of harnessing the capabilities of large language models (LLM). Given the inherent ambiguity present in natural language, it is intuitive to consider the possible advantages of prompting with less ambiguous prompt styles, like pseudo-code.

In this paper, we explore if prompting via pseudo-code instructions helps improve the performance of pre-trained language models. We manually create a dataset[1] of pseudo-code prompts for 132 different tasks spanning classification, QA, and generative language tasks, sourced from the Super-NaturalInstructions dataset (Wang et al., 2022b). Using these prompts along with their counterparts in natural language, we study their performance on two LLM families - BLOOM (Scao et al., 2023), CodeGen (Nijkamp et al., 2023). Our experiments show that using pseudo-code instructions leads to better results, with an average increase (absolute) of 7-16 points in F1 scores for classification tasks and an improvement (relative) of 12-38% in aggregate ROUGE-L scores across all tasks. We include detailed ablation studies which indicate that code comments, docstrings, and the structural clues encoded in pseudo-code all contribute towards the improvement in performance. To the best of our knowledge, our work is the first to demonstrate how pseudo-code prompts can be helpful in improving the performance of pre-trained LMs.

## 1 Introduction

Prompting with natural language instructions has recently emerged as a popular method of harnessing the capabilities of large language models. In addition to fine-tuning, models are often fine-tuned using instructions on a large collection of datasets

---

[*]Equal contribution

[1]Code and dataset available at https://github.com/mayank31398/pseudo-code-instructions

**Listing 1** An example pseudo-code instruction for the task from Wang et al. (2022b). A successful model is expected to use the provided pseudo-code instructions and output responses to a pool of evaluation instances.

```python
def generate_sentiment(sentence: str) -> str:
    """For the given sentence, the task is to
    predict the sentiment. For positive
    sentiment return "positive" else return
    "negative".

    Parameters:
        sentence (str): input sentence
    Returns:
        str: sentiment of the input
    """

    # predict the sentiment
    if sentiment_is_positive(sentence):
        return "positive"
    else:
        return "negative"

>>> generate_sentiment(
    "that has a charmingly bourbon air."
)
```

to help improve the ability of LMs to follow instructions and performance on unseen tasks (Wei et al., 2022a; Wang et al., 2022b).

However, natural language instructions can be ambiguous and under-specified, and therefore have multiple interpretations – including detailed instructions may not always be beneficial, as it can add to the complexity of reasoning for models. This has led to the growing body of work around 'prompt-engineering' where specialized prompting strategies are developed for different domains and task types (Zhao et al., 2021; Reynolds and McDonell, 2021; Arora et al., 2023; Liu et al., 2023; Zamfirescu-Pereira et al., 2023). In addition, inference-time prompting strategies that specifically aid multi-step reasoning have also been found to be helpful – e.g: the inclusion of chain-of-thought reasoning in few-shot settings results

in improved performance over standard prompts (Wei et al., 2022b), the infamous "*Let's think step-by-step*"-prompt for boosting 0-shot performance (Kojima et al., 2022).

---

**Algorithm 1** Attention Block

---
1: **function** TRANSFORMERS_ATTENTION_BLOCK($Q$, $K$, $V$)
2:     **Input:** $Q$, $K$, and $V$: input matrices.
3:     **Output:** The output of the attention block.
4:     $scores \leftarrow Q \cdot K^T$
5:     $attention\_weights \leftarrow softmax(scores)$
6:     $weighted\_values \leftarrow attention\_weights \cdot V$
7:     $output \leftarrow \sum_{i=1}^{n} weighted\_values_i$
8:     **return** $output$
9: **end function**

---

Given the inherent ambiguity present in natural language, it is intuitive to consider the advantages of prompting with less ambiguous prompt styles, such as the use of pseudo-code. Pseudo-code is an informal set of code-like constructs, which tend to be easy to interpret for humans but are not necessarily compilable/executable. They are often used to express complex ideas, processes, and flows – for example, Algorithm 1 expresses a summarized version of what happens within a Multi-Head Attention block (Vaswani et al., 2017) in pseudo-code. Arguably, expressing the same ideas in natural language could result in ambiguity and would perhaps require detailed text for clarity, which adds to the complexity.

In light of recent successes in NLP tasks achieved by code models (Madaan et al., 2022; Zhang et al., 2023a,b), this study aims to examine the efficacy of using pseudo-code instructions for prompting as a means of enhancing model performance. This study is driven by the hypothesis that using pseudo-code as prompts could offer a natural advantage to models in NLP tasks, owing to the concise and clearer expression of ideas in pseudo-code. To test the hypothesis that prompting large language models with pseudo-code instead of natural language data could be helpful, we created pseudo-code prompts[2] for 132 different tasks spanning 28 distinct task types, sourced from the Super-NaturalInstructions dataset (Wang et al., 2022b) (see Listing 1 for an example). Using these prompts along with their counterparts from natural language, we study their performance on two LLM families: BLOOM (Scao et al., 2023) and Code-Gen (Nijkamp et al., 2023). Both LLM families have been trained on natural language as well as code data.

We compare the performance of both styles of prompts on classification tasks, QA tasks, as well as a mix of other language generation tasks. Our experiments indicate that prompting with pseudo-code instructions indeed helps, and they result in an absolute gain of 7-16 points in F1 scores on classification tasks, and 12-38% relative improvement in aggregate ROUGE-L scores across all tasks.

**Contributions:** In summary, our paper makes the following contributions: (i) We release a dataset of 132 pseudo-code prompts spanning 28 different task types; (ii) Through a series of detailed experiments on two publicly available open-access LLM families, we demonstrate how prompting with pseudo-code instructions results in a marked improvement in performance over prompting with natural language instructions; (iii) We include detailed ablation studies indicating that code comments, docstrings, and the structural clues encoded in pseudo-code all contribute towards the improvement in performance.

To the best of our knowledge, our work is the first to demonstrate how pseudo-code prompts[3] can be helpful in improving the performance of pre-trained LMs. Our findings not only emphasize the significance of leveraging pseudo-code for prompting but also shed light on the specific elements within pseudo-code that contribute to the observed improvements.

## 2 Related Work

Finetuning large language models on instruction datasets can enhance their performance and even their ability to generalize to unseen tasks (Wei et al., 2021; Chung et al., 2022). Many aspects of instruction finetuning such as the number of tasks, model size, and finetuning on chain-of-thought data have been found to be useful (Chung et al., 2022). Consequently, significant efforts have been invested in manually creating instruction datasets, as well as using existing generative models to train and evaluate language models (Mishra et al., 2021; Bach et al., 2022; Wang et al., 2022b,a). The instructions available in instruction tuning datasets are mostly in natural language, but have been applied for both natural language tasks and programming tasks. But alternatives to natural language instructions such as programming language code, pseudo-code, symbols (MacCartney and Manning, 2007) etc. have

---

[2]The pseudo-code instructions for each of these tasks were created by the authors of this paper.

[3]In the rest of the paper, we use the words 'pseudo-code' and 'code' interchangeably when referring to prompts.

not been thoroughly explored even for programming tasks. Compared to natural language, code or pseudo-code has less ambiguity due to its inherent nature of using functions or steps that contribute towards accomplishing a task. This makes them a natural choice for specifying instructions. Recently, few works (MarvinAI; Madaan et al., 2022; Zhang et al., 2023a,b) have explored code and pseudo-code as inputs. Unlike contemporaneous work by Zhang et al. (2023a) we find that pseudo-code instructions indeed provide better performance over NL instructions on a wide variety of tasks.

## 3 Dataset

The Super-NaturalInstructions dataset (Wang et al., 2022b) comprises $1,616$ diverse NLP tasks, and each task contains the task instruction, positive/negative examples, and instances. We sampled a mixture of 132 tasks that did not require multilingual capabilities and re-wrote instructions for a subset of this dataset using Python constructs. Note that we borrow Python constructs only to express our prompts in pseudo-code and our prompts do not result in executable Python code. Further, we do not include any additional steps/instructions that were not present in the original natural language instructions.

All task instructions follow the schema as described in Listing 1. The schema consists of the following elements.

**Function Prototype:** This defines the prototype of the main pseudo-code function. The function names are descriptive and summarize the task to be performed. They also include all variables passed as input along with their data types and return type. We follow the PEP 8[4] style guidelines for writing the pseudo-code and use strongly typed prototypes. We avoid declaring global variables whenever possible and pass them as arguments to a method. To the extent possible, we also avoid the use of classes and enumerations. Line number 1 in Listing 1 provides an example function prototype for a sentiment classification task.

**DocString:** The docstring provides detailed instructions on the task to be performed in natural language. Often, this is a paraphrased version of the original natural language instruction. The docstring ends with a list of parameters (with their types) being passed and the return type from the

---

[4]https://peps.python.org/pep-0008/

function. An example docstring for the sentiment classification task is presented in line numbers 2 to 13 in Listing 1.

**Function Definition:** This includes the bulk of the pseudo-code instruction describing how to solve the particular task. To the extent possible, the function definitions do not leave out any information contained in the docstring. Pseudo-code in the function definition are written as sub-task functions. These sub-task functions are usually not defined and often use descriptive names, arguments and variables. We include in-line comments indicating what is accomplished by the sub-task function and the role of the arguments if required. We sometimes also define secondary sub-task functions if it requires additional details or if the descriptive function name may not be adequate to specify the goal of the sub-task function. We assume the availability of basic helper functions such as concat_str, search etc., and do not include any import statements.

Line numbers 14 to 17 present function definition for sentiment classification task. The function calls sentiment_is_positive sub-task function which checks if the sentiment of the given sentence is positive or not. This function is not explicitly defined in the instruction.

**Pre-processor:** Since the pseudo-code instructions expect inputs as arguments, we need to parse the inputs provided in the Super-NaturalInstructions dataset (Wang et al., 2022b) (which provides pre-formatted inputs). For each pseudo-code instruction, we also include an executable python pre-processor which is used for parsing the input.

### 3.1 Dataset Statistics

We created instructions for 132 tasks that have instructions and input/output pairs in English language. We group the tasks into three classes: Classification Tasks (Table 1), QA tasks (Table 2) and other language generation tasks (Table 3). These tasks cover a total of 28 different categories and span 72 unique datasets. For each task we sample 1000 instances for evaluation.

## 4 Evaluation

In order to study if instruction specification via pseudo-code results in improved performance over

baseline NL English instructions, we choose to experiment with BLOOM (Scao et al., 2023), Code-Gen (Nijkamp et al., 2023) models. Our choice of models is motivated by the fact that these models have not been instruction-fine-tuned on the Natural Instructions dataset. In addition, they have both been trained on code and natural language data.

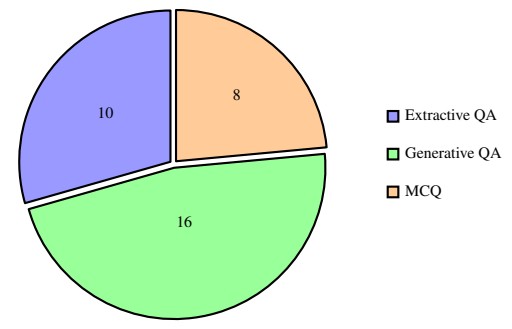

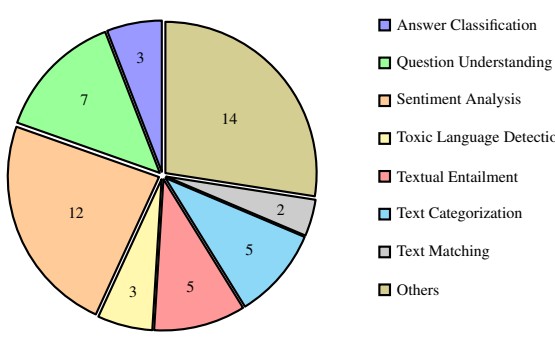

| Task Category | Datasets |
|---|---|
| Answer Classification | MultiRC (Khashabi et al., 2018), McTaco (Ben Zhou and Roth, 2019), TWEETQA (Xiong et al., 2019) |
| Answer Verification | MultiRC (Khashabi et al., 2018) |
| Commonsense Classification | ATOMIC (Sap et al., 2019) |
| Coreference Selection | Numeric Fused-Head (Elazar and Goldberg, 2019) |
| Dialogue Selection | SPOLIN (Cho and May, 2020), DSTC3 (Henderson et al., 2014) |
| Grammar Error Detection | CoLA (Warstadt et al., 2019) |
| Intent Identification | DailyDialog (Li et al., 2017) |
| Irony Detection | SemEval2018-Task3 (Van Hee et al., 2018) |
| Linguistic Classification | SentEval (Conneau and Kiela, 2018) |
| Prime Number Classification | Synthetic (Wang et al., 2022b) |
| Program Execution | Synthetic (Wang et al., 2022b) |
| Question Understanding | McTaco (Ben Zhou and Roth, 2019), DROP (Dua et al., 2019), TREC (Li and Roth, 2002), DREAM (Sun et al., 2019), FreebaseQA (Jiang et al., 2019) |
| Section Classification | CODA-19 (Huang et al., 2020) |
| Sentiment Analysis | The Multilingual Amazon Reviews Corpus (Keung et al., 2020), Sentiment140 (Go et al., 2009), SST-2 (Socher et al., 2013), PerSenT (Bastan et al., 2020), Amazon Review Polarity (Face), PEC (Zhong et al., 2020), Poem Sentiment (Sheng and Uthus, 2020) |
| Text Categorization | MultiNLI (Williams et al., 2018), DDO (Durmus and Cardie, 2019), SemEval-2020 Task 7 (Hossain et al., 2020), Scruples (Lourie et al., 2021) |
| Text Matching | AFS (Misra et al., 2016), PAWS (Zhang et al., 2019) |
| Text Quality Classification | McTaco (Ben Zhou and Roth, 2019) |
| Textual Entailment | MultiNLI (Williams et al., 2018), SNLI (Bowman et al., 2015), e-SNLI (Camburu et al., 2018), Defeasible-NLI (Rudinger et al., 2020), ATOMIC (Sap et al., 2019) |
| Toxic Language Detection | CAD (Vidgen et al., 2021), Jigsaw (cjadams et al., 2019), Hate Speech Offensive (Davidson et al., 2017) |
| Wrong Candidate Generation | McTaco (Ben Zhou and Roth, 2019) |

Table 1: Collection of classification tasks used in our work

The BLOOM models are trained on the ROOTS corpus (Laurençon et al., 2022) consisting of 46 natural and 13 programming languages. On the other hand, the CodeGen models are trained on the Pile corpus (Gao et al., 2020), Google's publicly available BigQuery and BigPython datasets (Nijkamp et al., 2023). The BLOOM models have been trained on a mixture of natural language and code simultaneously. As for the CodeGen models we utilize, they were initially trained on natural language and subsequently received additional

| Task Category | Datasets |
|---|---|
| Extractive QA | ROPES (Lin et al., 2019a), Odd-Man-Out (Stanovsky and Hopkins, 2018), SQuAD1.1 (Rajpurkar et al., 2016), Synthetic (Wang et al., 2022b), MCScript (Ostermann et al., 2018), PICO (Jin and Szolovits, 2018), MWSC (McCann et al., 2019), OPUS (Tiedemann, 2012), CoQA (Reddy et al., 2019) |
| Generative QA | Quoref (Dasigi et al., 2019), McTaco (Ben Zhou and Roth, 2019), DROP (Dua et al., 2019), MultiRC (Khashabi et al., 2018), PIQA (Bisk et al., 2020), Synthetic (Wang et al., 2022b), BREAK (Wolfson et al., 2020), Natural Questions (Kwiatkowski et al., 2019), AmbigQA (Min et al., 2020), CoQA (Reddy et al., 2019), TriviaQA (Joshi et al., 2017) |
| MCQ | Essential, QuaRel (Tafjord et al., 2018), WinoGrande (Sakaguchi et al., 2021), MultiNLI (Williams et al., 2018), ReCoRD (Zhang et al., 2018), MMMLU (Hendrycks et al., 2021) |

Table 2: Collection of QA tasks used in our work

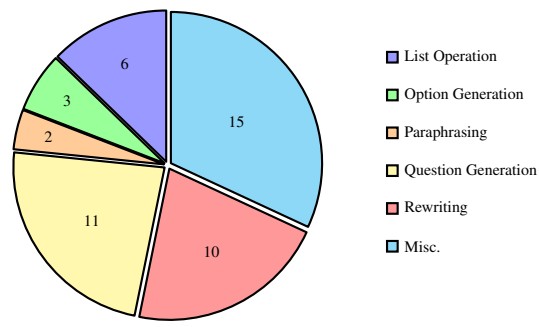

| Task Category | Datasets |
|---|---|
| List Operation | CoNaLa (Yin et al., 2018), Synthetic (Tiedemann, 2012), Youtube Caption Corrections (2dot71mily) |
| Option Generation | aNLI (Nie et al., 2020), ASSET (Alva-Manchego et al., 2020), ROCStories (Mostafazadeh et al., 2017) |
| Paraphrasing | ZEST (Weller et al., 2020), PARANMT-50M (Wieting and Gimpel, 2018) |
| Question Generation | CosmosQA (Huang et al., 2019), WinoGrande (Sakaguchi et al., 2021), ROPES (Lin et al., 2019b), SQuAD1.1 (Rajpurkar et al., 2016), StrategyQA (Geva et al., 2021), SQuAD2.0 (Rajpurkar et al., 2018), BoolQ (Clark et al., 2019), CoQA (Reddy et al., 2019), QA-ZRE (Levy et al., 2017) |
| Rewriting | WinoGrande (Sakaguchi et al., 2021), aNLI (Nie et al., 2020), ASSET (Alva-Manchego et al., 2020), ZEST (Weller et al., 2020), SNLI (Bowman et al., 2015) |
| Misc. | DROP (Dua et al., 2019), WinoGrande (Sakaguchi et al., 2021), QASC (Khot et al., 2020), Essential (Khashabi et al., 2017), ROPES (Lin et al., 2019a), StoryCloze (Mostafazadeh et al., 2016), Country Barcode Prefix dataset, Country Region in World dataset, Gigaword (Graff et al., 2003), GAP (Webster et al., 2018), SPOLIN (Cho and May, 2020), XL-WiC (Raganato et al., 2020) |

Table 3: Collection of language generation tasks used in our work

training focused specifically on Python code.

Our choice of models allows us to setup a controlled environment where we can study the impact of prompting in natural language and pseudo-code.

Most recent instruction-tuned models have either seen the Super-NaturalInstructions dataset (Wang et al., 2022b) in some form (Longpre et al., 2023) or they do not have tokenizers that will meaningfully process code syntax (Raffel et al., 2020), and therefore can not be used in our study. By empirically studying the performance of models on these prompts, we hope to inform future work on training an instruction-tuned model using pseudo-code instructions.

## 4.1 Model Configurations

For all of the experiments conducted in this paper, we use BLOOM-3B, BLOOM 7B (Scao et al., 2023), CodeGen-mono 2B, and CodeGen-mono 6B (Nijkamp et al., 2023) models. The inference was performed using A100 80 GB GPUs. To accelerate the inference of all models, we utilized DeepSpeed-Inference (Aminabadi et al., 2022) in fp16, which resulted in an average inference throughput improvement of around 1.7x, compared to the standard HuggingFace (Wolf et al., 2020) inference. We used greedy decoding for all our experiments for reproducibility and restricted generated outputs to 100 tokens. Even for classification tasks, we generate the class labels using auto-regressive decoding instead of picking the class label with lowest perplexity. This is done because not all class labels can be mapped to a single token for all tasks. This technique of evaluating performance of classification tasks is often employed when using closed LLMs, such as those behind APIs (eg: OpenAI's GPT4 (OpenAI, 2023), Google's PaLM (Chowdhery et al., 2022) etc).

## 4.2 Metrics

We adopt different metrics for each task-category: we measure the performance of classification tasks using micro, macro and weighted F1 scores, and for QA and language generation tasks we use the ROUGE-L metric. We report the ROUGE-L, Exact Match (EM), and ANLS - Average Normalized Levenshtein Similarity (Biten et al., 2019) for all tasks.

## 4.3 Output post-processing

Since the models we experiment with have not been fine-tuned for instruction following, they tend to generate excess text after the output for the given task. We therefore post-process the outputs to ensure models are not penalized in our evaluation due to excess generations. We post-process

all outputs by truncating by the newline character '\n'. Furthermore, the output is subjected to additional post-processing, including punctuation removal and lower casing.

## 4.4 Results

Through our experiments we aim to answer the following questions: (i) What is the difference in performance between prompting pre-trained language and code models with pseudo-code prompts versus natural language prompts? (ii) How does increasing model size affect the efficacy of pseudo-code prompts? (iii) To what extent does structured prompting, such as the use of function names, docstrings, inline comments, and arguments, impact performance on tasks?

### 4.4.1 Prompting with Pseudo-code

Table 4 compares the performance of prompting with pseudo-code (referred to as code instructions) and natural language instructions in 0-shot settings. Results have been grouped by model family and size.

As can be seen, for all model families and sizes, prompting with pseudo-code results in a significant improvement in performance. The performance on classification tasks is especially notable, for example, the gains on weighted F1 vary between 7-16 F1 points (absolute). Furthermore, the relative performance improvement on all other tasks, as measured by ROUGE-L, varies between 12-38%. The overall performance as measured by ROUGE-L, ANLS and Exact Match also report similar trends.

**Comparison of CodeGen vs BLOOM** Despite most tasks being non-code tasks, CodeGen, a model designed for code applications, outperforms BLOOM models, even when using natural language instructions (see metrics for 'All Tasks'). Similar behavior has been anecdotally reported (Fu and Khot, 2022; Madaan et al., 2022), but has possibly not been investigated using as many tasks as presented in this paper. Note, however, that using pseudo-code prompts in the code models results in better performance than any other prompt-model configuration.

**Performance on QA tasks** Interestingly, we find that on QA tasks, the performance of pseudo-code instructions is better than natural-language instructions, when using the CodeGen model. However, this is not the case when using BLOOM.

| Model | Instruction Format | Classification Tasks | | | QA Tasks | Generation tasks | All Tasks | | |
|---|---|---|---|---|---|---|---|---|---|
| | | Macro F1 | Micro F1 | Weighted F1 | ROUGE-L | ROUGE-L | ROUGE-L | ANLS | EM |
| Majority Class | | **0.296** | **0.509** | **0.362** | - | - | - | - | - |
| CodeGen 2B | Code Instructions | **0.272** | **0.417** | **0.354** | **0.175** | **0.317** | **0.330** | **0.261** | **0.202** |
| | NL Instructions | 0.068 | 0.306 | 0.239 | 0.154 | 0.254 | 0.265 | 0.195 | 0.147 |
| CodeGen 6B | Code Instructions | **0.311** | **0.443** | **0.375** | **0.201** | **0.327** | **0.354** | **0.283** | **0.218** |
| | NL Instructions | 0.052 | 0.278 | 0.215 | 0.132 | 0.271 | 0.257 | 0.187 | 0.134 |
| BLOOM 3B | Code Instructions | **0.116** | **0.351** | **0.288** | 0.147 | **0.271** | **0.279** | **0.215** | **0.165** |
| | NL Instructions | 0.082 | 0.275 | 0.214 | **0.159** | 0.234 | 0.250 | 0.180 | 0.132 |
| BLOOM 7B | Code Instructions | **0.174** | **0.369** | **0.285** | 0.150 | **0.298** | **0.297** | **0.232** | **0.176** |
| | NL Instructions | 0.046 | 0.247 | 0.203 | **0.156** | 0.276 | 0.247 | 0.172 | 0.122 |

Table 4: Performance of models when prompted using pseudo-code instructions and natural language instructions in 0-shot settings. (i) In each model, prompting with pseudo-code instructions results in much higher performance in almost all the tasks (ii) For each model family, increasing scale helps improve performance (iii) Prompting CodeGen (a model designed for code) results in better performance than BLOOM. (iv) Prompting BLOOM models with Natural Language instructions instead of code-instructions results in higher performance on QA tasks.

| | CodeGen 6B | | | | | | BLOOM 7B | | | | | |
|---|---|---|---|---|---|---|---|---|---|---|---|---|
| | Code Instructions | | | NL Instructions | | | Code Instructions | | | NL Instructions | | |
| QA Task | EM | ROUGE-L | ANLS | EM | ROUGE-L | ANLS | EM | ROUGE-L | ANLS | EM | ROUGE-L | ANLS |
| **Extractive QA** | 0.140 | 0.303 | 0.189 | 0.045 | 0.188 | 0.077 | 0.047 | 0.184 | 0.077 | 0.047 | 0.227 | 0.086 |
| **Generative QA** | 0.045 | 0.129 | 0.068 | 0.029 | 0.095 | 0.045 | 0.028 | 0.101 | 0.042 | 0.032 | 0.115 | 0.047 |
| **MCQ** | 0.196 | 0.213 | 0.210 | 0.082 | 0.106 | 0.083 | 0.184 | 0.201 | 0.197 | 0.107 | 0.143 | 0.108 |

Table 5: 0-shot performance of CodeGen 6B and BLOOM 7B models on QA tasks from our dataset. As can be seen, pseudo-code instructions applied on the CodeGen model results in the best overall performance on all categories of QA tasks. However, comparing the performance of Natural Language Instructions, we find that it performs marginally better than pseudo-code instructions on non-MCQ QA tasks when using the BLOOM 7B model.

We investigated this further and observed that for most QA tasks, the instructions in pseudo-code are not significantly more detailed or easier to understand than natural-language instructions. As an example, the pseudo-code instruction for answer generation from the SQuAD dataset merely contains the following statement in its function definition: `return get_answer_from_passage(passage, question)` and reflects the details included in the natural instructions.

We further analysed the results across QA task categories and found that pseudo-code instructions always help with multiple-choice questions (MCQ) tasks (see Table 5 for a comparison between CodeGen 6B and BLOOM 7B). We believe that this is because, understanding the instructions in such tasks may be more involved. For illustration, instructions in MCQ tasks often include details about *how* answers are expected – eg: "*choose the correct option A, B, C*", "*Select Option 1 - Value 1, Option 2 - Value 2*". Depending on the instructions, the models may be required to return options, values, or both which adds a degree of complexity to the instructions as compared to other types of QA.

The discrepancy in performance between CodeGen and BLOOM on QA tasks (see Table 5), could be attributed to the fact that the structure from code prompts could be better leveraged by code models as programming languages and aspects of code syntax (structure) are likely to be better represented in a code model such as CodeGen. This brings us to our next question – What is the contribution of structure that may be present in prompts?

### 4.4.2 Contribution of Structure in prompts

The reasons behind the performance improvement when using pseudo-code prompts are likely to be a combination of factors, including the use of descriptive function names that convey the function's purpose (such as `get_answer(question)`), a model that can effectively utilize structured information, and a structured prompt for a task that could further benefit from few-shot examples.

We therefore experiment with different structured prompting styles and report their results in Table 6. We study the performance of CodeGen and

| Model | Instruction Format | Classification Tasks | | | QA Tasks | Generation Tasks | All Tasks | | |
|---|---|---|---|---|---|---|---|---|---|
| | | Macro F1 | Micro F1 | Weighted F1 | ROUGE-L | ROUGE-L | ROUGE-L | ANLS | EM |
| CodeGen 2B | Code Instructions (0) | **0.272** | **0.417** | **0.354** | 0.175 | **0.317** | **0.330** | **0.262** | **0.202** |
| | Function Declaration (0) | 0.159 | 0.079 | 0.085 | 0.124 | 0.252 | 0.153 | 0.083 | 0.043 |
| | Function Declaration (2) | 0.105 | 0.267 | 0.257 | **0.185** | 0.294 | 0.256 | 0.188 | 0.137 |
| | Function Invocation (2) | 0.097 | 0.253 | 0.238 | 0.183 | 0.296 | 0.251 | 0.183 | 0.131 |
| | Generic Function Invocation (2) | 0.064 | 0.282 | 0.244 | 0.167 | 0.257 | 0.245 | 0.185 | 0.131 |
| | NL Examples (2) | 0.003 | 0.005 | 0.007 | 0.081 | 0.126 | 0.069 | 0.017 | 0.006 |
| CodeGen 6B | Code Instructions (0) | **0.311** | **0.444** | **0.375** | **0.201** | **0.327** | **0.354** | **0.283** | **0.218** |
| | Function Declaration (0) | 0.019 | 0.101 | 0.109 | 0.162 | 0.273 | 0.179 | 0.111 | 0.063 |
| | Function Declaration (2) | 0.134 | 0.309 | 0.281 | 0.196 | 0.299 | 0.281 | 0.212 | 0.154 |
| | Function Invocation (2) | 0.133 | 0.296 | 0.269 | 0.192 | 0.302 | 0.275 | 0.208 | 0.149 |
| | Generic Function Invocation (2) | 0.062 | 0.244 | 0.215 | 0.167 | 0.262 | 0.239 | 0.175 | 0.121 |
| | NL Examples (2) | 0.000 | 0.000 | 0.001 | 0.102 | 0.168 | 0.088 | 0.023 | 0.006 |
| BLOOM 3B | Code Instructions (0) | **0.116** | **0.351** | **0.288** | 0.147 | **0.271** | **0.279** | **0.214** | **0.165** |
| | Function Declaration (0) | 0.000 | 0.014 | 0.016 | 0.108 | 0.229 | 0.116 | 0.054 | 0.015 |
| | Function Declaration (2) | 0.080 | 0.237 | 0.217 | **0.164** | 0.249 | 0.225 | 0.159 | 0.115 |
| | Function Invocation (2) | 0.073 | 0.227 | 0.211 | **0.164** | 0.234 | 0.215 | 0.149 | 0.107 |
| | Generic Function Invocation (2) | 0.032 | 0.173 | 0.168 | 0.161 | 0.246 | 0.203 | 0.137 | 0.086 |
| | NL Examples (2) | 0.000 | 0.025 | 0.031 | 0.150 | 0.208 | 0.122 | 0.056 | 0.024 |
| BLOOM 7B | Code Instructions (0) | **0.174** | **0.369** | **0.285** | 0.150 | **0.298** | **0.297** | **0.232** | **0.176** |
| | Function Declaration (0) | 0.004 | 0.021 | 0.027 | 0.111 | 0.242 | 0.124 | 0.058 | 0.017 |
| | Function Declaration (2) | 0.072 | 0.256 | 0.227 | **0.191** | 0.289 | 0.257 | 0.182 | 0.128 |
| | Function Invocation (2) | 0.086 | 0.248 | 0.221 | 0.189 | 0.286 | 0.250 | 0.176 | 0.123 |
| | Generic Function Invocation (2) | 0.039 | 0.199 | 0.178 | 0.187 | 0.276 | 0.232 | 0.155 | 0.097 |
| | NL Examples (2) | 0.000 | 0.009 | 0.009 | 0.132 | 0.182 | 0.106 | 0.038 | 0.016 |

Table 6: Study of structured prompts: Performance of models when prompted using 0-shot pseudo-code instructions, function declaration in 0-shot and 2-shot settings as well as 2-shot prompting with a 'generic' function name and the use of only examples. The number N in the brackets indicates N-shot prompt. (i) Except for the performance on QA tasks, in each model, prompting with pseudo-code instructions results in much higher performance which indicates that detailed instructions are helpful (ii) For each model family, and prompting style, increasing model scale improves performance (iii) As before, prompting a model designed for code, CodeGen, results in better performance than BLOOM.

BLOOM with five types of prompts: (i) Pseudo-code instructions, (ii) Prompts that make use of function declaration (declare function name only), (iii) a structured prompt consisting only of task examples in 2-shot settings using the task-descriptive function name (iv) a structured prompt consisting only of task examples in 2-shot settings using a generic function name – 'func' (v) using the Natural Language examples (without instructions) in 2-shot settings. Details about each prompt have been included in the Appendix.

We make three important observations from Table 6. First, code-instructions in 0-shot settings consistently yield the best overall performance compared to other structured prompts. Second, on average, the CodeGen model consistently outperforms BLOOM on all tasks. Lastly, the QA tasks in our dataset, which are relatively easy to express in natural language instructions, also benefit from structured prompts, particularly when prompted with examples.

It can be inferred from these observations that the performance gains resulting from the use of pseudo-code prompts are likely due to clearer task instructions, and not just the exploitation of superfluous patterns from in-context learning. These findings reinforce the results from the previous experiment, which showed that code models are more capable of exploiting structured prompts. In the case of QA tasks in our dataset, it is worth noting that since the pseudo-code instructions are not as detailed, even utilizing a simpler structured prompt with examples can significantly enhance performance as compared to natural language prompts.

### 4.4.3 Impact of pseudo-code documentation

In this section, we study the contribution of comments and docstrings present in our pseudo-code instructions towards the improvement in performance. We first study the performance of pseudo-code prompts with and without the use of docstrings and code comments.

As can be seen in Table 7, the inclusion of comments as well as the docstring in the pseudo-code instruction prompt helps improve performance. This indicates that not only is the structure of the prompts being exploited by the model, the models are also relying on additional helper text present in the documentation. We, therefore, also investigate if the use of these elements from pseudo-code could also benefit natural language instruction prompts.

The lower half of table 7 studies the performance of natural-language prompts with and without the use of pseudo-code comments and docstring. We

| Model | Instruction Format | Classification Tasks | | | QA Tasks | Generation Tasks | All Tasks | | |
|---|---|---|---|---|---|---|---|---|---|
| | | Macro F1 | Micro F1 | Weighted F1 | ROUGE-L | ROUGE-L | ROUGE-L | ANLS | EM |
| CodeGen 6B | Code Instructions | **0.311** | **0.444** | **0.375** | **0.201** | **0.327** | **0.354** | **0.283** | **0.218** |
| | Code Instructions without docstrings and comments | 0.263 | 0.409 | 0.348 | 0.195 | 0.327 | 0.335 | 0.266 | 0.201 |
| BLOOM 7B | Code Instructions | **0.174** | **0.369** | **0.285** | **0.150** | **0.298** | **0.297** | **0.232** | **0.176** |
| | Code Instructions without docstrings and comments | 0.145 | 0.316 | 0.247 | 0.144 | 0.291 | 0.269 | 0.204 | 0.151 |
| CodeGen 6B | NL Instructions | 0.052 | 0.278 | 0.215 | 0.132 | 0.271 | 0.257 | 0.187 | 0.134 |
| | NL Instructions with docstrings and comments | **0.062** | **0.312** | **0.254** | **0.139** | **0.293** | **0.275** | **0.208** | **0.148** |
| BLOOM 7B | NL Instructions | **0.046** | 0.247 | 0.203 | 0.156 | **0.276** | 0.247 | 0.172 | 0.122 |
| | NL Instructions with docstrings and comments | 0.044 | **0.303** | **0.233** | **0.165** | 0.263 | **0.266** | **0.199** | **0.147** |

Table 7: Ablation: Zero-Shot Setting. (i) In each model, prompting with pseudo-code instructions results in much higher performance on QA and classification tasks (ii) For each model family, increasing scale helps improve performance (iii) As before, prompting a model designed for code, CodeGen results in better performance than BLOOM. On average, in the CodeGen model, the use of code comments and docstrings helps improve the performance of natural language prompts. However, it appears for BLOOM, only the larger-sized model is able to consistently use the additional details in the prompt to improve performance.

find that the performance of natural language instructions also improves by the inclusion of comments and docstring for each model family and configuration. We hypothesize that the gains may be attributable to a form of step-by-step reasoning derived from pseudo-code comments especially in complex tasks.

## 4.5 Summary of findings

We now summarize our findings for easy reference.
**Effect of Prompting Style:** From Table 4 we observe that 0-shot prompting of pre-trained models with pseudo-code prompts results in better performance than natural language prompts. This is true for both code models and language models. The gains are more pronounced for the code models.
**Effect of Structure in prompts:** Pseudo-code prompts include many elements such as the function declaration, docstring, comments etc. From Table 6 we find that while information from the function declaration, and a task-indicative function name help, using the complete pseudo-code prompt is most useful.

Further, from Table 7 we find that the pseudo-code instruction still works better than any prompt created with natural language instructions, even when docstring and comments from pseudo-code are included in the natural language instruction. This suggests the gains from prompting in pseudo-code are not just due to comments and docstrings (which could help reinforce the task instructions), but also due to clearer instructions in pseudo-code.
**Effect of Model Size:** From Table 4 we find that

in 0-shot settings, with the increase in scale, the performance of pseudo-code instructions improves for both model families. However, when using natural language instructions, this is not the case. We hypothesize, that since none of these models are instruction-tuned, larger scales exacerbate the propensity of the models being primed for language completion.
**Code vs. Natural Language models:** We find that code models are better suited for exploiting pseudo-code prompts compared to language models. As can be seen from Table 4 (see metrics for 'All Tasks'), the use of natural language instructions on CodeGen results in better performance than their use on BLOOM.

## 5 Conclusion and Future Work

In this paper we presented our work on prompting with pseudo-code instructions. We created a collection of pseudo-code instructions comprising of 132 NLP tasks from the Super-NaturalInstructions dataset (Wang et al., 2022b). We evaluated the performance of the following families of models - CodeGen and BLOOM at different model sizes and found that prompting all models with pseudo-code instructions results in significant gains as compared to prompting with NL instructions. Our work opens up multiple directions of future work. It is interesting to observe that not only do pseudo-code instructions help when used with code models, they also work better on models designed for natural language tasks. In addition, the fact that code mod-

els used in our experiments perform better than NL models, even when prompted with natural language instructions, suggests that it could be useful to explore instruction tuning of code models instead of pure NL models for NL applications. Based on the findings of this paper it may also be useful to consider the effects of instruction fine-tuning with pseudo-code instructions as opposed to NL instructions.

Another aspect worth studying is how traditional chain-of-thought may compare with pseudo-code prompts – how would reasoning enabled by pseudo-code instructions compare with chain-of-thought reasoning with and without fine-tuning? Further, pseudo-code instructions may not only be used as direct inputs to a model, but they could also be used to create intermediate responses that a model needs to generate prior to returning a response.

## Limitations

Our results have been reported on two model families – CodeGen and BLOOM at scales of 2-7B parameters. It remains to be seen if our findings would hold at larger model sizes. It is possible that better reasoning enabled by larger model sizes could reduce the benefit of prompting with pseudo-code instructions but we have not investigated this in our work. In addition, our work does not include any multi-lingual NLP tasks – BLOOM was specifically trained to be able to support multiple languages and it is possible this model design choice could play a role in our findings when we compare code (CodeGen) and NL (BLOOM) models against each other. Moreover, both models have been trained on different datasets and this also affects the intrinsic reasoning capabilities of these models. Lastly, and importantly, the use of pseudo-code for prompting LLMs is limited by the expectation that it requires technical expertise to write them, thus reducing their widespread usage.

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

## A   Appendix

### A.1   Results on an additional LLMs

We also perform experiments using the recently released, Falcon-7B (Almazrouei et al., 2023) model. The results are presented in Table 8 and also demonstrates improved performance with pseudo-code prompts.

### A.2   Pseudo-Code Validation

To ensure that the pseudo-code instructions follow the guidelines provided, we run an automatic test. The test code calls the `preprocess` function defined for each example from the Super-NaturalInstructions dataset (Wang et al., 2022b) for that task. The returned values from the `preprocess` function are compared against the arguments in the function prototype. Any mismatch in the data type or the number of arguments results in error. The instruction creator is given feedback to correct the errors.

#### A.2.1   Prompt Styles

In this section, we describe the various prompting styles used to study the effect of pseudo-code vs NL prompting. Here, we show a simple task to generate the sentiment of a given sentence. This is task 833 in Super-NaturalInstructions dataset.

#### A.2.2   Prompting with Pseudo-code instructions

For the pseudo-code prompting, we use the instructions that are created by the authors of this paper. The pseudo-code instructions have a much richer structure than natural language instructions and are more elaborate and simple to understand. They contain docstrings, return types and might also contain comments, function invocations etc. For preparing the few shot examples and the input query, we treat the example as a python interpreter running in the linux terminal and use the special markers '>>>' for the input. We don't use any special markers for the outputs. An example for 0-shot and 2-shot shot prompting is shown in Listings 2 and 3 respectively.

We also measure the impact of removing the docstrings and comments from the code instruction. An example for 0-shot and 2-shot prompting is shown in Listings 4 and 5 respectively.

**Listing 2** Code instructions (0-shot prompt) for sentiment classification task

```python
def generate_sentiment(sentence: str) -> str:
    """For the given sentence, the task is to
    predict the sentiment. For positive sentiment
    return "positive" else return "negative".

    Parameters:
        sentence (str): input sentence
    Returns:
        str: sentiment of the input
    """

    # predict the sentiment
    if sentiment_is_positive(sentence):
        return "positive"
    else:
        return "negative"

>>> generate_sentiment(
    "that has a charmingly bourbon air."
)
```

**Listing 3** Code instructions (2-shot prompt) for sentiment classification task

```python
def generate_sentiment(sentence: str) -> str:
    """For the given sentence, the task is to
    predict the sentiment. For positive sentiment
    return "positive" else return "negative".

    Parameters:
        sentence (str): input sentence
    Returns:
        str: sentiment of the input
    """

    # predict the sentiment
    if sentiment_is_positive(sentence):
        return "positive"
    else:
        return "negative"

>>> generate_sentiment(
    "tormented by the quickened blood of the "
    "roots"
)
"negative"

>>> generate_sentiment(
    "radiant as moses from the mount, he stood"
)
"positive"

>>> generate_sentiment(
    "that has a charmingly bourbon air."
)
```

#### A.2.3   Prompting with function prototype

We try prompting the models with function prototypes with all docstrings, comments and code logic removed from the base pseudo-code instruction. The function prototype instructions are composed of the function names, arguments and their types

| Model | Instruction Format | Classification Tasks | | | QA Tasks | Generation tasks | All Tasks | | |
|---|---|---|---|---|---|---|---|---|---|
| | | Macro F1 | Micro F1 | Weighted F1 | ROUGE-L | ROUGE-L | ROUGE-L | ANLS | EM |
| Majority Class | | **0.296** | **0.509** | **0.362** | - | - | - | - | - |
| Falcon 7B | Code Instructions | **0.068** | **0.339** | **0.259** | 0.152 | 0.265 | **0.275** | **0.207** | **0.161** |
| | NL Instructions | 0.017 | 0.206 | 0.197 | **0.172** | **0.273** | 0.242 | 0.149 | 0.102 |

Table 8: Performance of models when prompted using pseudo-code instructions and natural language instructions in 0-shot settings. (i) In each model, prompting with pseudo-code instructions results in much higher performance in almost all the tasks

**Listing 4** Code instructions without docstrings and comments (0-shot prompt) for sentiment classification task

```
def generate_sentiment(sentence: str) -> str:
    if sentiment_is_positive(sentence):
        return "positive"
    else:
        return "negative"

>>> generate_sentiment(
    "that has a charmingly bourbon air."
)
```

**Listing 5** Code instructions without docstrings and comments (2-shot prompt) for sentiment classification task

```
def generate_sentiment(sentence: str) -> str:
    if sentiment_is_positive(sentence):
        return "positive"
    else:
        return "negative"

>>> generate_sentiment(
    "tormented by the quickened blood of the "
    "roots"
)
"negative"

>>> generate_sentiment(
    "radiant as moses from the mount, he stood"
)
"positive"

>>> generate_sentiment(
    "that has a charmingly bourbon air."
)
```

and the return types. This method of prompting is devoid of any pseudo-code. An example for 0-shot and 2-shot prompting is shown in Listings 6 and 7 respectively.

### A.2.4 Prompting with NL instructions

For natural language prompts, we use the original instructions provided as part of the Super-NaturalInstructions dataset (Wang et al., 2022b). For natural language instruction prompting, we

**Listing 6** Function prototype (0-shot prompt) for sentiment classification task

```
def generate_sentiment(sentence: str) -> str:

>>> generate_sentiment(
    "that has a charmingly bourbon air."
)
```

**Listing 7** Function prototype (2-shot prompt) for sentiment classification task

```
def generate_sentiment(sentence: str) -> str:

>>> generate_sentiment(
    "tormented by the quickened blood of the "
    "roots"
)
"negative"

>>> generate_sentiment(
    "radiant as moses from the mount, he stood"
)
"positive"

>>> generate_sentiment(
    "that has a charmingly bourbon air."
)
```

use the prompts provided as part of the Super-NaturalInstructions dataset without any modification. We add special 'input:' and 'output:' markers in the few shot examples and the input query to the model as shown in Listings 8 and 9.

**Listing 8** Natural instructions (0-shot prompt) for sentiment classification task

```
In this task, you need to identify the sentiment
of the given sentence as one of "positive" or
"negative".

input: that has a charmingly bourbon air.
output:
```

**Listing 9** Natural instructions (2-shot prompt) for sentiment classification task

```
In this task, you need to identify the sentiment
of the given sentence as one of "positive" or
"negative".

input: tormented by the quickened blood of the
roots
output: negative

input: radiant as moses from the mount, he stood
output: positive

input: that has a charmingly bourbon air.
output:
```

### A.2.5 Prompting with NL instructions and NL comments from the pseudo-code

We also try experimenting by adding the docstrings and comments to the NL instructions from the Super-NaturalInstructions dataset (Wang et al., 2022b) as shown in the example in Listings 10 and 11.

**Listing 10** Natural instructions with docstrings (0-shot prompt) for sentiment classification task

```
In this task, you need to identify the sentiment
of the given sentence as one of "positive" or
"negative".

"""For the given sentence, the task is to
predict the sentiment. For positive sentiment
return "positive" else return "negative".

Parameters:
    sentence (str): input sentence
Returns:
    str: sentiment of the input
"""

# predict the sentiment

input: that has a charmingly bourbon air.
output:
```

### A.2.6 Prompting without instructions

We also study the effect of prompting without instructions. We try this method of prompting in three settings:

1. Function Invocation (refer Listings 12 and 13)

2. Generic Invocation (refer Listings 14 and 15)

3. Natural Language examples (refer Listings 16 and 17)

**Listing 11** Natural instructions with docstrings (2-shot prompt) for sentiment classification task

```
In this task, you need to identify the sentiment
of the given sentence as one of "positive" or
"negative".

"""For the given sentence, the task is to
predict the sentiment. For positive sentiment
return "positive" else return "negative".

Parameters:
    sentence (str): input sentence
Returns:
    str: sentiment of the input
"""

# predict the sentiment

input: tormented by the quickened blood of the
roots
output: negative

input: radiant as moses from the mount, he stood
output: positive

input: that has a charmingly bourbon air.
output:
```

**Listing 12** Function invocation (0-shot prompt) for sentiment classification task

```
>>> generate_sentiment(
    "that has a charmingly bourbon air."
)
```

**Listing 13** Function invocation (2-shot prompt) for sentiment classification task

```
>>> generate_sentiment(
    "tormented by the quickened blood of the "
    "roots"
)
"negative"

>>> generate_sentiment(
    "radiant as moses from the mount, he stood"
)
"positive"

>>> generate_sentiment(
    "that has a charmingly bourbon air."
)
```

**Listing 14** Generic function invocation (0-shot prompt) for sentiment classification task

```
>>> func(
    "that has a charmingly bourbon air."
)
```

| Model | Instruction Format | Classification Tasks | | | QA Tasks | Generation Tasks | All Tasks | | |
|---|---|---|---|---|---|---|---|---|---|
| | | Macro F1 | Micro F1 | Weighted F1 | ROUGE-L | ROUGE-L | ROUGE-L | ANLS | EM |
| CodeGen 2B | Code Instructions | **0.137** | **0.295** | **0.272** | **0.187** | **0.299** | **0.269** | **0.202** | **0.148** |
| | NL Instructions | 0.000 | 0.004 | 0.006 | 0.082 | 0.130 | 0.071 | 0.017 | 0.006 |
| CodeGen 6B | Code Instructions | **0.145** | **0.317** | **0.292** | **0.194** | **0.304** | **0.285** | **0.219** | **0.159** |
| | NL Instructions | 0.000 | 0.001 | 0.002 | 0.101 | 0.172 | 0.089 | 0.024 | 0.006 |
| BLOOM 3B | Code Instructions | **0.086** | **0.254** | **0.227** | **0.151** | **0.248** | **0.226** | **0.164** | **0.121** |
| | NL Instructions | 0.005 | 0.060 | 0.060 | 0.151 | 0.207 | 0.140 | 0.070 | 0.038 |
| BLOOM 7B | Code Instructions | **0.072** | **0.250** | **0.227** | **0.191** | **0.279** | **0.250** | **0.176** | **0.124** |
| | NL Instructions | 0.000 | 0.120 | 0.014 | 0.137 | 0.186 | 0.109 | 0.041 | 0.018 |

Table 9: Performance with 2-shot prompts. (i) In each model, prompting with pseudo-code instructions results in much higher performance (ii) For each model family, increasing scale helps improve performance (iii) As before, prompting a model designed for code, CodeGen results in better performance than BLOOM. (iv) Surprisingly, as compared to 0-shot prompting (Table 4), there is a marked drop in performance for all model configurations and all tasks, except in QA tasks, where there is an improvement in performance.

---

**Listing 15** Generic function invocation (2-shot prompt) for sentiment classification task

```
>>> func(
    "tormented by the quickened blood of the "
    "roots"
)
"negative"

>>> func(
    "radiant as moses from the mount, he stood"
)
"positive"

>>> func(
    "that has a charmingly bourbon air."
)
```

---

**Listing 16** Natural examples (0-shot prompt) for sentiment classification task

```
input: that has a charmingly bourbon air.
output:
```

---

**Listing 17** Natural examples (2-shot prompt) for sentiment classification task

```
input: tormented by the quickened blood of the
roots
output: negative

input: radiant as moses from the mount, he stood
output: positive

input: that has a charmingly bourbon air.
output:
```

---

## A.3 2-shot Prompting with Pseudo-code instructions

Given that structured prompts, such as those based on function declarations, benefit from 2-shot prompts, we investigate whether the performance of pseudo-code prompts can be further improved with 2-shot prompts. Table 9 reports the performance of both families of models - CodeGen and BLOOM when using pseudo-code prompts and natural language instruction prompts in 2-shot settings.

Interestingly we find that, as compared to the results reported in Table 4 the performance of each corresponding model-prompt configuration is lower than its 0-shot counterpart. While this may appear surprising, similar findings have been reported in prior work (Reynolds and McDonell, 2021; Zhang et al., 2023a). Perhaps the performance in few-shot settings could improve with additional examples, but we do not experiment with more than 2-shot settings due to limitations imposed by the size of input context length available to models.

After a study of outputs generated by the models in 2-shot settings, we observe that in many cases, in the absence of extensive task-specific prompt-engineering and output processing, models are likely to generate additional continuation examples instead of solving the task. The fact that the pseudo-code prompts perform better indicate that models seem to "*interpret*" the instructions better in this form.

## A.4 Ablation Experiments

As can be seen in Table 10 and 11, the inclusion of comments as well as the docstring in the pseudo-code instruction prompt and natural language instructions helps improve performance for smaller models too.

| Model | Instruction Format | Classification Tasks | | | QA Tasks | Generation Tasks | All Tasks | | |
|---|---|---|---|---|---|---|---|---|---|
| | | Macro F1 | Micro F1 | Weighted F1 | ROUGE-L | ROUGE-L | ROUGE-L | ANLS | EM |
| CodeGen 2B | NL Instructions | 0.068 | 0.306 | 0.239 | **0.154** | 0.254 | 0.265 | 0.195 | 0.147 |
| | NL Instructions with docstrings and comments | **0.098** | **0.349** | **0.270** | 0.136 | **0.258** | **0.275** | **0.208** | **0.161** |
| CodeGen 6B | NL Instructions | 0.052 | 0.278 | 0.215 | 0.132 | 0.271 | 0.257 | 0.187 | 0.134 |
| | NL Instructions with docstrings and comments | **0.062** | **0.312** | **0.254** | **0.139** | **0.293** | **0.275** | **0.208** | **0.148** |
| BLOOM 3B | NL Instructions | **0.082** | **0.275** | **0.214** | **0.159** | **0.234** | **0.250** | **0.180** | **0.132** |
| | NL Instructions with docstrings and comments | 0.046 | 0.233 | 0.209 | 0.121 | 0.202 | 0.213 | 0.146 | 0.111 |
| BLOOM 7B | NL Instructions | **0.046** | 0.247 | 0.203 | 0.156 | **0.276** | 0.247 | 0.172 | 0.122 |
| | NL Instructions with docstrings and comments | 0.044 | **0.303** | **0.233** | **0.165** | 0.263 | **0.266** | **0.199** | **0.147** |

Table 10: Ablation: On average, in the CodeGen model the use of code comments and docstrings in 0-shot setting helps improve performance of natural language prompts. However, it appears on BLOOM, only the larger sized model is able to consistently use the additional details in the prompt to improve performance.

| Model | Instruction Format | Classification Tasks | | | QA Tasks | Generation Tasks | All Tasks | | |
|---|---|---|---|---|---|---|---|---|---|
| | | Macro F1 | Micro F1 | Weighted F1 | ROUGE-L | ROUGE-L | ROUGE-L | ANLS | EM |
| CodeGen 2B | Code Instructions | **0.272** | **0.417** | **0.354** | **0.175** | **0.317** | **0.330** | **0.262** | **0.202** |
| | Code Instructions without docstrings and comments | 0.241 | 0.389 | 0.337 | 0.159 | 0.305 | 0.309 | 0.241 | 0.185 |
| CodeGen 6B | Code Instructions | **0.311** | **0.444** | **0.375** | **0.201** | **0.327** | **0.354** | **0.283** | **0.218** |
| | Code Instructions without docstrings and comments | 0.263 | 0.409 | 0.348 | 0.195 | 0.327 | 0.335 | 0.266 | 0.201 |
| BLOOM 3B | Code Instructions | **0.116** | **0.351** | **0.288** | **0.147** | **0.271** | **0.279** | **0.215** | 0.165 |
| | Code Instructions without docstrings and comments | 0.094 | 0.302 | 0.249 | 0.132 | 0.259 | 0.248 | 0.117 | 0.183 |
| BLOOM 7B | Code Instructions | **0.174** | **0.369** | **0.285** | **0.150** | **0.298** | **0.297** | **0.232** | **0.176** |
| | Code Instructions without docstrings and comments | 0.145 | 0.316 | 0.247 | 0.144 | 0.291 | 0.269 | 0.204 | 0.151 |

Table 11: Ablation: Using 0-shot code instructions without docstrings and comments (i) In each model, prompting with pseudo-code instructions results in much higher performance on QA and classification tasks (ii) For each model family, increasing scale helps improve performance (iii) As before, prompting a model designed for code, CodeGen results in better performance than BLOOM.