# OpenReview forum: "Prompting with Pseudo-Code Instructions"
_EMNLP/2023/Conference — EMNLP 2023 Main_

### Official Review · Reviewer_FjEd · 2023-07-31

**Soundness:** 3

**Excitement:**

4: Strong: This paper deepens the understanding of some phenomenon or lowers the barriers to an existing research direction.

**Paper Topic And Main Contributions:**

This paper focused on using pseudo-code as instructions when prompting large language models (LLMs), instead of the traditional approach which used natural language prompts. The authors stated that using pseudo-code as instructions is much better than natural language under a zero-shot prompting setting.

Contributions:
1. The authors proposed to use pseudo-code as instructions instead of traditional natural language instructions.
2. They provided pseudo-code instructions on 132 tasks in the Super Natural Instructions benchmark.
3. Experimental results on two LLM families (BLOOM & CodeGen) proved the advantage of using pseudo-code instructions over natural language instructions under the zero-shot setting. The authors also showed the importance of using docstrings, comments and function declarations.

**Questions For The Authors:**

A. I am somewhat confused about the wording of "structure" in section 4.4.2. Does "structure" mean "the pseudo-code is a structured type of input, compared to unstructured natural language"?

B. How are the 2-shot experiments conducted for natural language instructions in table 8? Can you give an example of the prompt? It is weird that 2-shot NL instructions are far worse than 0-shot NL instructions.

**Reasons To Accept:**

1. The idea of leveraging pseudo-code is novel. The motivation is that pseudo-code are less ambiguous, more concise and clearer than natural language instructions. The motivation is clear and innovative.
2. The authors provided a set of pseudo-code instructions on 132 tasks, which could be beneficial for future research.
3. The authors proved the advantage of pseudo-code instructions in both code-based LLM (CodeGen) and text-based LLM (BLOOM).
4. The authors analyzed the components in pseudo-code instructions and highlighted the critical role of task-indicative function names, docstrings and comments.

**Reasons To Reject:**

1. The fairness of the experiments is somewhat questionable. In general, vanilla LLMs (i.e., LLMs that are not instruction-tuned) are not good zero-shot learners. Particularly, they tend to generate excessive outputs because they are basically doing language completion instead of solving classification/QA problems. Given the significant improvements in the experiments (e.g., table 4) after using pseudo-code instructions, I am wondering whether this is because in some tasks, pseudo-code instructions have more detailed specifications or constraints on the format of the output. As far as I know, the specification of output format in the instruction may significantly impact the zero-shot performance of a vanilla LLM.
2. Another setting that I found questionable is 2-shot prompting. In section 4.4.2, the authors mentioned using 2-shot natural language examples without any natural language instructions on vanilla LLMs. Why are instructions not used? I think a consensus in LLMs is that few-shot instruction-based prompting is better than zero-shot prompting, at least this should be the case for the natural language instructions. However, in the reported results, 2-shot natural language prompting is far worse than its 0-shot counterpart.
3. This paper lacks examples of (1) pseudo-code instructions, and (2) illustrations of different prompts used in section 4.4.2 (it is hard to imagine the difference between these ablation prompts without any examples). The only example is figure 1, but there is no comparison between the pseudo-code instruction and the natural language instruction.

**Reproducibility:**

4: Could mostly reproduce the results, but there may be some variation because of sample variance or minor variations in their interpretation of the protocol or method.

**Reviewer Confidence:**

4: Quite sure. I tried to check the important points carefully. It's unlikely, though conceivable, that I missed something that should affect my ratings.

---

> ### Author Rebuttal · Authors · 2023-08-28
>
> We thank the reviewer for their feedback and we are pleased to note that the reviewer found our work innovative.
>
> ---
>
> ### **Question 1**
>
> The fairness of the experiments is somewhat questionable. In general, vanilla LLMs (i.e., LLMs that are not instruction-tuned) are not good zero-shot learners. Particularly, they tend to generate excessive outputs because they are basically doing language completion instead of solving classification/QA problems. Given the significant improvements in the experiments (e.g., table 4) after using pseudo-code instructions, I am wondering whether this is because in some tasks, pseudo-code instructions have more detailed specifications or constraints on the format of the output. As far as I know, the specification of output format in the instruction may significantly impact the zero-shot performance of a vanilla LLM.
>
> ### **Answer 1**
>
> We agree it would indeed be interesting to assess the performance of instruction-tuned models  -- unfortunately, most instruction tuned models have seen SuperNatInstructions or versions of it and creating new instruction datasets and their pseudo-code prompts is beyond the scope of this work. The input format and the output formats of prompts can indeed affect zero-shot performance -- we include samples of prompts and all our other experimental settings for reference in this rebuttal.
>
> For all prompts, we use new-line character as the delimiter and extract the text till the first occurrence of new line character as the generated text to ensure models prone to excessive text generation are not unfairly penalized.
>
> ---
>
> ### **Question 2**
>
> Another setting that I found questionable is 2-shot prompting. In section 4.4.2, the authors mentioned using 2-shot natural language examples without any natural language instructions on vanilla LLMs. Why are instructions not used? I think a consensus in LLMs is that few-shot instruction-based prompting is better than zero-shot prompting, at least this should be the case for the natural language instructions. However, in the reported results, 2-shot natural language prompting is far worse than its 0-shot counterpart.
>
> ### **Answer 2**
>
> Our goal in Section 4.4.2 was to specifically assess the contribution made by structure in the input prompts -- for pseudo code prompts we have many elements to experiment with (function declaration, definition, with and without two-shot examples, etc). But for NL text, we could only consider an ablation where the instruction is removed and the structured 2-shot example is retained.
>
> Further, we would like to highlight that we indeed experimented with full two-shot prompting as well  --  Please see Table 8 in the appendix of the paper. The reviewer has correctly highlighted that the lower performance of two-shot is a bit surprising - however as stated in lines 1298-1301 of the paper, lower performance with few-shot as compared to zero-shot has been noted by other works as well.
>
> ---
>
> ### **Question 3**
>
> This paper lacks examples of (1) pseudo-code instructions, and (2) illustrations of different prompts used in section 4.4.2 (it is hard to imagine the difference between these ablation prompts without any examples). The only example is figure 1, but there is no comparison between the pseudo-code instruction and the natural language instruction.
>
> ### **Answer 3**
>
> We apologize for the oversight --- we will update the paper with the examples.
>
> **Example 1 (Pseudo-code instruction)**:
> ```python
> def generate_sentiment(tweet: str) -> str:
>     """Given a text from tweet, the task in get the sentiment about the tweet. For a positive sentiment tweet return "positive" else return "negative".
>     Parameters:
>         tweet: text from tweet
>         Returns:
>                 str:answer to the question
>     """
>     # predict the sentiment (positive or negative)
>     if get_sentiment(tweet) == "positive":
>         return "positive"
>     else:
>         return "negative"
>
> >>> generate_sentiment("its sad that the rats are becoming aggressive against the guinea pigs. I have to seperate them. ")
> ```
>
> **Example 2 (Pseudo-code instruction with 2-shot examples)**:
> ```python
> def generate_sentiment(tweet: str) -> str:
>     """Given a text from tweet, the task in get the sentiment about the tweet. For a positive sentiment tweet return "positive" else return "negative".
>     Parameters:
>         tweet: text from tweet
>         Returns:
>                 str:answer to the question
>     """
>     # predict the sentiment (positive or negative)
>     if get_sentiment(tweet) == "positive":
>         return "positive"
>     else:
>         return "negative"
>
> >>> generate_sentiment("tormented by the quickened blood of rots")
> "negative"
>
> >>> generate_sentiment("radiant as moses from the mount, he stood")
> "positive"
>
> >>> generate_sentiment("its sad that the rats are becoming aggressive against the guinea pigs. I have to seperate them. ")
> ```
>
> **Example 3 (Pseudo-code instruction with no docstring)**:
>  ```python
> def generate_sentiment(tweet: str) -> str:
>     # predict the sentiment (postive or ne
>     if get_sentiment(tweet) == "positive":
>         return "positive"
>     else:
>         return "negative"
>
> >>> generate_sentiment("its sad that the rats are becoming aggressive against the guinea pigs. I have to seperate them. ")
> ```
>
> **Example 4 (Pseudo-code instruction with no docstring 2-shot examples)**:
> ```python
> def generate_sentiment(tweet: str) -> str:
>     # predict the sentiment (positive or negative)
>     if get_sentiment(tweet) == "positive":
>         return "positive"
>     else:
>         return "negative"
>
> >>> generate_sentiment("tormented by the quickened blood of rots")
> "negative"
>
> >>> generate_sentiment("radiant as moses from the mount, he stood")
> "positive"
>
> >>> generate_sentiment("its sad that the rats are becoming aggressive against the guinea pigs. I have to seperate them. ")
> ```
>
> **Example 5 (Pseudo-code instruction with function definition only)**:
>  ```python
> def generate_sentiment(tweet: str) -> str:
>
> >>> generate_sentiment("its sad that the rats are becoming aggressive against the guinea pigs. I have to seperate them. ")
> ```
>
> **Example 6 (Pseudo-code instruction with function definition only and 2-shot examples)**:
> ```python
> def generate_sentiment(tweet: str) -> str:
>
> >>> generate_sentiment("tormented by the quickened blood of rots")
> "negative"
>
> >>> generate_sentiment("radiant as moses from the mount, he stood")
> "positive"
>
> >>> generate_sentiment("its sad that the rats are becoming aggressive against the guinea pigs. I have to seperate them. ")
> ```
>
> **Example 7 (Natural instruction)**:
> ```text
> In this task, you need to identify the sentiment of the given sentence as one of 'positive' or 'negative'.
>
> Input: its sad that the rats are becoming aggressive against the guinea pigs. I have to seperate them.
> Output:
> ```
>
> **Example 8 (Natural instruction with 2-shot examples)**:
> ```text
> In this task, you need to identify the sentiment of the given sentence as one of 'positive' or 'negative'.
>
> Input: tormented by the quickened blood of rots
> Output: negative
>
> Input: radiant as moses from the mount, he stood
> Output: positive
>
> Input: its sad that the rats are becoming aggressive against the guinea pigs. I have to seperate them.
> Output:
> ```
>
> **Example 9 (Natural instruction with docstring)**:
> ```text
> In this task, you need to identify the sentiment of the given sentence as one of 'positive' or 'negative'.
>     """Given a text from tweet, the task in get the sentiment about the tweet. For a positive sentiment tweet return "positive" else return "negative".
>     Parameters:
>         tweet: text from tweet
>         Returns:
>                 str:answer to the question
>     """
>
> Input: its sad that the rats are becoming aggressive against the guinea pigs. I have to seperate them.
> Output:
> ```
>
> **Example 10 (Natural instruction with docstring and 2-shot examples)**:
> ```text
> In this task, you need to identify the sentiment of the given sentence as one of 'positive' or 'negative'.
>     """Given a text from tweet, the task in get the sentiment about the tweet. For a positive sentiment tweet return "positive" else return "negative".
>     Parameters:
>         tweet: text from tweet
>         Returns:
>                 str:answer to the question
>     """
>
> Input: tormented by the quickened blood of rots
> Output: negative
>
> Input: radiant as moses from the mount, he stood
> Output: positive
>
> Input: its sad that the rats are becoming aggressive against the guinea pigs. I have to seperate them.
> Output:
> ```
>
> **Example 11 (No instruction in pseudocode)**:
> ```text
> >>> generate_sentiment("its sad that the rats are becoming aggressive against the guinea pigs. I have to seperate them. ")
> ```
>
> **Example 12 (No instruction in pseudocode and 2-shot examples)**:
> ```text
> >>> generate_sentiment("tormented by the quickened blood of rots")
> negative
>
> >>> generate_sentiment("radiant as moses from the mount, he stood")
> positive
>
> >>> generate_sentiment("its sad that the rats are becoming aggressive against the guinea pigs. I have to seperate them. ")
> ```
>
> **Example 13 (No instruction in pseudocode and generic function call)**:
> ```text
> >>> func("its sad that the rats are becoming aggressive against the guinea pigs. I have to seperate them. ")
> ```
>
> ---
>
> ### **Question 4**
>
> I am somewhat confused about the wording of "structure" in section 4.4.2. Does "structure" mean "the pseudo-code is a structured type of input, compared to unstructured natural language"?
>
> ### **Answer 4**
>
> Yes, by structure, we refer to programming language syntax. This can be seen in the examples listed above.
>
> ---
>
> ### **Question 5**
>
> How are the 2-shot experiments conducted for natural language instructions in table 8? Can you give an example of the prompt? It is weird that 2-shot NL instructions are far worse than 0-shot NL instructions.
>
> ### **Answer 5**
>
> Please refer to example 8 above for 2-shot prompting with NL instructions.

---

### Official Review · Reviewer_wLJ1 · 2023-08-07

**Soundness:** 4

**Excitement:**

4: Strong: This paper deepens the understanding of some phenomenon or lowers the barriers to an existing research direction.

**Paper Topic And Main Contributions:**

The paper proposed to use pseudo-code as instructions of LLMs instead of natural language instructions. It created a collection of pseudo-code instructions comprising of 132 NLP tasks. Experiment are done with two LLMs: CodeGen and BLOOM. Experimental results show that pseudo-code instructions can improve task performance over natural language instructions on most tasks. It also did many ablation studies and analysis on model size and prompt structures.

**Reasons To Accept:**

1. An interesting attempt to replace natural language prompts with more structured code-based prompts. This can benefit many practical tasks.
2. Promising experimental results and solid analysis on a large collection of tasks.
3. The collected dataset can be beneficial for the NLP community.

**Reasons To Reject:**

1. The experimented LLMs are not very recent models. I understand that the area is moving forward rapidly, but it would be much more interesting if the paper can also experiment with recent models with larger size like LLAMA.
2. Experiments tasks are all from Super-NaturalInstructions, which is still very different from current popular LLM tasks like planning and open generation.

**Reproducibility:**

4: Could mostly reproduce the results, but there may be some variation because of sample variance or minor variations in their interpretation of the protocol or method.

**Reviewer Confidence:**

4: Quite sure. I tried to check the important points carefully. It's unlikely, though conceivable, that I missed something that should affect my ratings.

---

> ### Author Rebuttal · Authors · 2023-08-28
>
> We thank the reviewer for their feedback and for rating our paper string and exciting! We are pleased that the reviewer found our experiments insightful.
>
> We include a response to some points raised by the reviewer.
>
> ---
>
> ### **Question 1**
>
> The experimented LLMs are not very recent models. I understand that the area is moving forward rapidly, but it would be much more interesting if the paper can also experiment with recent models with larger size like LLAMA.
>
> ### **Answer 1**
>
> Indeed, the field is moving rapidly -  to address the concern about recency, we conducted an experiment with a relatively recent model such as Falcon-7B.  As can be seen in the table below, we see a similar trend where the model prompted with pseudo-code instructions outperforms model prompted with natural language instruction.
>
> |    Model   | Instruction  Format | Classification Tasks |          | QA Tasks | Generation tasks | All Tasks |
> |:----------:|:-------------------:|:--------------------:|:--------:|:--------:|:----------------:|:---------:|
> |            |                     | Macro F1             | Micro F1 | ROUGE-L  | ROUGE-L          | ROUGE-L   |
> | Falcon-7b  | Code Instructions   | 0.068                | 0.339    | 0.152    | 0.265            | 0.275     |
> |            | NL Instructions     | 0.017                | 0.206    | 0.172    | 0.273            | 0.242     |
>
> ---
>
> ### **Question 2**
>
> Experiments tasks are all from Super-NaturalInstructions, which is still very different from current popular LLM tasks like planning and open generation.
>
> ### **Answer 2**
>
> As noted in the paper we present our experiments on a wide variety of tasks including classification and QA and other language generation tasks. The goal of our experiments was to study if prompting in pseudo-code could help improve the "instruction-following" abilities of models. As our experiments suggest, prompting in pseudo code instructions does indeed help models and will therefore also benefit when used in LLM Prompt chains for larger applications where LLMs are allowed to influence decisions. We are unsure if the reviewer meant something else by "planning" -- we'd be happy to further clarify during the discussion period.

---

### Official Review · Reviewer_ZMgi · 2023-08-08

**Soundness:** 4

**Excitement:**

4: Strong: This paper deepens the understanding of some phenomenon or lowers the barriers to an existing research direction.

**Paper Topic And Main Contributions:**

The paper discusses the recent trend of using natural language instructions to harness the capabilities of large language models. Given the ambiguity in natural language, the authors explore the potential advantages of using less ambiguous prompt styles, specifically pseudo-code. They manually create a dataset of pseudo-code prompts for 132 tasks, which include classification, QA, and generative language tasks, sourced from the Super-NaturalInstructions dataset. The performance of these prompts is compared against their natural language counterparts using two LLM families: BLOOM and CodeGen. Results indicate that pseudo-code instructions lead to better outcomes, with an average increase of 7-16 points in F1 scores for classification tasks and a 12-38% improvement in aggregate ROUGE-L scores across all tasks. The study also delves into the factors contributing to this improvement, such as code comments, docstrings, and structural cues in pseudo-code.

**Reasons To Accept:**

The authors released a dataset containing 132 pseudo-code prompts spanning 28 different task types.

Pseudo-code instructions significantly improve performance compared to natural language instructions.

Detailed ablation studies highlight the elements within pseudo-code that contribute to the observed improvements.


**Reasons To Reject:**

The results have been reported on two model families – CodeGen and BLOOM at scales of 2-7B parameters. It remains uncertain if the findings would hold at larger model sizes.

**Reproducibility:**

4: Could mostly reproduce the results, but there may be some variation because of sample variance or minor variations in their interpretation of the protocol or method.

**Reviewer Confidence:**

4: Quite sure. I tried to check the important points carefully. It's unlikely, though conceivable, that I missed something that should affect my ratings.

---

> ### Author Rebuttal · Authors · 2023-08-28
>
> We thank the reviewer for their feedback and for rating our paper strong and exciting. We include a response to a point raised by the reviewer.
>
> ---
>
> ### **Question 1**
>
> The results have been reported on two model families – CodeGen and BLOOM at scales of 2-7B parameters. It remains uncertain if the findings would hold at larger model sizes.
>
> ### **Answer 1**
>
> We acknowledge that the trends could be different at larger parameter scales and we leave that to future work. Due to the large number number of tasks for evaluation running a larger model was computationally expensive, so we limited our evaluations to model size from 2-7B parameters. However, to help assuage concerns about generalization we include experiment on another model family - Falcon-7B. As can be seen in the table below, we see a similar trend where the model prompted with pseudo-code instructions outperforms model prompted with natural language instruction.
>
> |    Model   | Instruction  Format | Classification Tasks |          | QA Tasks | Generation tasks | All Tasks |
> |:----------:|:-------------------:|:--------------------:|:--------:|:--------:|:----------------:|:---------:|
> |            |                     | Macro F1             | Micro F1 | ROUGE-L  | ROUGE-L          | ROUGE-L   |
> | Falcon-7b  | Code Instructions   | 0.068                | 0.339    | 0.152    | 0.265            | 0.275     |
> |            | NL Instructions     | 0.017                | 0.206    | 0.172    | 0.273            | 0.242     |

---

### Official Review · Reviewer_45A7 · 2023-08-11

**Soundness:** 3

**Excitement:**

4: Strong: This paper deepens the understanding of some phenomenon or lowers the barriers to an existing research direction.

**Paper Topic And Main Contributions:**

The paper introduces a new idea of prompting large language models with pseudo-code instructions. Compared with natural language, pseudo-code is often more specific in describing the task. Following this idea, the authors construct a new evaluation dataset by rewriting the natural language instructions into pseudo-code for 132 tasks from Super-NaturalInstructions. Experiments show that pseudo-code prompts lead to better performance when evaluated under zero-shot setting with models that have not been instruction-finetuned. Further analysis demonstrates that pseudo-code prompts are more powerful with models specifically trained on code.

**Questions For The Authors:**

1. How did you calculate the classification F1, and what's the random-guess baseline for classification tasks in Table 4? The macro F1 scores for NL instructions seem very low. Did you look at the token-level probability distribution, or did you simply measure the lexical overlap between the greedy output and the groundtruth?
2. I suspect that one advantage of pseudo-code instruction is to clearly define the output space for classification. For example, the model is expected to output "Yes/No" instead of "True/False" for a binary problem, but the NL instruction doesn't make it clear. I wonder how often this happens.
3. It would be helpful to include more complicated examples in appendix than Figure 1. For example, a pseudo-code instruction with step-by-step comments as mentioned in L454-457 will be very illustrative.

**Reasons To Accept:**

1. The idea of prompting with pseudo-code is novel and interesting. It inspires many future directions for exploration, such as pseudo-code prompt engineering/generation and instruction finetuning with pseudo-code.
2. Pseudo-code prompting exhibits better performance than natural language instructions for LLMs pretrained on code.
3. Extensive analysis is conducted to show the contribution of each component in pseudo-code prompting.

**Reasons To Reject:**

I don't have a clear reason to reject this paper.

**Reproducibility:**

5: Could easily reproduce the results.

**Reviewer Confidence:**

3: Pretty sure, but there's a chance I missed something. Although I have a good feel for this area in general, I did not carefully check the paper's details, e.g., the math, experimental design, or novelty.

---

> ### Author Rebuttal · Authors · 2023-08-28
>
> We thank the reviewer for their feedback. We are pleased to note that the reviewer found our work novel and interesting!
> We include a response to the questions below:
>
> ---
>
> ### **Question 1**
>
> How did you calculate the classification F1, and what's the random-guess baseline for classification tasks in Table 4? The macro F1 scores for NL instructions seem very low. Did you look at the token-level probability distribution, or did you simply measure the lexical overlap between the greedy output and the ground truth?
>
> ### **Answer 1**
>
> We extract the predicted answer from the generated output. We use new-line character as the delimiter and extract the text till the first occurrence of new line character as the generated text. This generated text was compared with the ground truth to calculate the F1 scores. We use exact match for computing F1-scores. The Rouge-L scores in column ALL includes the results on classification task too where we measure the Rouge score between the generated output and the ground truth.
>
> We didn't report classification performance based on token-level probability (or perplexity) of the class label in the main paper.
> However, to address the reviewer's question, we compute classification performance based on perplexity. The micro-F1 numbers for BLOOM-7B and CodeGen-6B are reported in the table below. We also report a random baseline (picking a class label at random) averaged over 5 different seeds. It can be clearly seen that pseudo-code instructions outperform the NL instructions by 8 points (approx.) on BLOOM-7B and 6 points (approx.) on Codegen-6B.
>
> |  | Pseudo-code instructions | Natural Language instructions | Random baseline (independent of model) |
> |---|---|---|---|
> | BLOOM-7B | **0.227** | 0.145 | 0.075 |
> | CodeGen-6B mono | **0.209** | 0.148 | 0.075 |
>
> ---
>
> ### **Question 2**
>
> I suspect that one advantage of pseudo-code instruction is to clearly define the output space for classification. For example, the model is expected to output "Yes/No" instead of "True/False" for a binary problem, but the NL instruction doesn't make it clear. I wonder how often this happens.
>
> ### **Answer 2**
>
> The output space is defined in both cases (pseudo-code and NL instructions). For example, in the following task we have 2 classes (Inversion & Original) for which we demonstrate both pseudo-code and NL instruction:
>
> **Pseudo-code instruction**:
> ```python
> def is_words_flipped(sentence: str) -> str:
>     """
>     Given a sentence, check if the sentence contains two consecutive words which needs to be flipped to make the sentence grammatical and meaningful. If yes, then return 'Inversion'. If no changes are required return 'Original'
>
>     Parameters:
>         sentence (str): a given sentence
>
>     Returns:
>         str: \"Inversion"\ or \"Original\"
>     """
>
>     # the functions checks if the sentence contains two words which needs to be flipped to make the sentence grammatical and meaningful
>     if words_need_to_be_flipped(sentence):
>         return "Inversion"
>     else:
>         return "Original"
> ```
>
> **NL instruction**:
> ```text
> In this task you are given a sentence. You must judge whether there exist two consecutive words within the sentence with flipped orders, that is, whether the sentence will make sense and be correct if the order of two consecutive words changes. Label the instances as \"Inversion\" or \"Original\" based on your judgment.
> ```
> As can be seen in both cases, the class labels are included as part of the instructions.
>
> ---
>
> ### **Question 3**
>
> It would be helpful to include more complicated examples in appendix than Figure 1. For example, a pseudo-code instruction with step-by-step comments as mentioned in L454-457 will be very illustrative.
>
> ### **Answer 3**
>
> We will include the examples in the accepted version of the paper. Here are a few included examples for pseudo-code instructions:
>
> **Example 1**:
> ```python
> def create_fill_in_the_blanks_question(context_word: str, mention_person1: str, mention_person2: str, target_blank: str, child_friendly_mode: bool = True) -> str:
>     """creates a fill in the blank question based on the mentioned persons, context word and the target person to be blanked"""
>
>     # initialize gender options
>     gender = ["male", "female", "trans"]
>     # choose gender to be used for PersonX and PersonY
>     gender_index = random(0, size(gender))
>     # select gender
>     selected_gender = gender[gender_index]
>     # get sentence with context word and mentioned persons based on count of mentions.
>     generated_sentence = get_sentence(
>         context_word,
>         child_friendly_mode,
>         mention_person1,
>         mention_person2,
>         mention_person_1_count=2,
>         mention_person_2_count=1,
>         min_word_length=15,
>         max_word_length=30,
>         person_gender=selected_index,
>     )
>     # blank second mention of mention_person1
>     blanked_sentence = blank_sentence(generated_sentence, blank_token="_", blank_target=mention_person1, blank_mention_index=2)
>     # return blanked sentence
>     return blanked_sentence
> ```
>
> **Example 2**:
> ```python
> def question_category_classification(sentence: str, question: str, category: str) -> str:
>     """
>     Given a sentence and a question around the sentence. A reasoning category is also provided. The task is to indicate if the question belongs to the provided reasoning vategory or not.
>
>     a) Event Duration: understanding of how events last
>     b) transient v. stationary: understanding of whether an event will change over time or not
>     c) event ordering: understanding of how events are usually ordered in nature
>     d) absolute timepoint: understanding of when events usually happen
>     e) frequency: how often an event is likely to be repeated
>
>     Parameters:
>         sentence (str): sentence
>         question (str): question
>         category (str): category
>     Returns:
>         str: `Yes.` or `No.`
>     """
>
>     if "Event Duration" in category:
>         # this function checks if the question has an understanding of how long the events last
>         if does_question_require_event_duration_reasoning(question, category, sentence):
>             return "Yes."
>         else:
>             return "No."
>     elif "transient v. stationary" in category:
>         # this function checks if the question has an understanding of whether an event will change over time or not
>         if does_question_require_event_temporal_reasoning(question, category, sentence):
>             return "Yes."
>         else:
>             return "No."
>     elif "event ordering" in category:
>         # this function checks if the question has an understanding of how events are ordered in nature
>         if does_question_require_event_ordering_reasoning(question, category, sentence):
>             return "Yes."
>         else:
>             return "No."
>     elif "absolute timepoint" in category:
>         # this function checks if the question has an understanding of when events usually happen
>         if does_question_require_event_happen_reasoning(question, category, sentence):
>             return "Yes."
>         else:
>             return "No."
>     elif "frequency" in category:
>         # this function checks if the question has an understanding of how often an event is likely to be repeated
>         if does_question_require_event_frequency_reasoning(question, category, sentence):
>             return "Yes."
>         else:
>             return "No."
> ```

---

### Meta-Review · Area_Chair_ooew · 2023-09-16

**Recommendation:** 5

**Metareview:**

This paper investigates if you can write prompts for LLMs in pseudocode rather than natural language and investigates how well this works. The authors discussion period had a spirited discussion with the authors and the reviewers where the reviewers' concerns were mostly resolved; this discussion shows that the results are interesting and surprising and left the reviewers convinced that the results were indeed sound.

---

### Decision · Program_Chairs · 2023-10-07

**Decision:**

Accept-Main

**Comment:**

This paper investigates if you can write prompts for LLMs in pseudocode rather than natural language and investigates how well this works. The authors discussion period had a spirited discussion with the authors and the reviewers where the reviewers' concerns were mostly resolved; this discussion shows that the results are interesting and surprising and left the reviewers convinced that the results were indeed sound.